# Examining the overlap in lymphatic filariasis prevalence and malaria insecticide-treated net access-use in endemic Africa

Joanna L. Whisnant[1], Mustafa Kamal Sikder[2], Gizachew Taddesse Akalu[3,4], Tsegaye Alemu[5,6], Mubarek Yesse Ashemo[7,8], Amelia Bertozzi-Villa[9], Annie J. Browne[10], Ewerton Cousin[1,11], Paulina Agnieszka Dzianach[12], Yalemzewod Assefa Gelaw[13,14], Peter W. Gething[15,12], Taren M. Gorman[1], Simon I. Hay[1,11], Olayinka Stephen Ilesanmi[16,17], Cathleen Keller[1], Juniper Boroka Kiss[18], Jailos Lubinda[19], Michael A. McPhail[18], Olivia D. Nesbit[1], Gideon Olamilekan Oluwatunase[20,21], Verner N. Orish[22,23], Amel Ouyahia[24,25], Susan Fred Rumisha[18,26], Adam Saddler[19], Afeez Abolarinwa Salami[27,28], Francesca Sanna[12], Desalegn Shiferaw[29,30], Jacques Lukenze Tamuzi[31,32], Daniel J. Weiss[33,12], Naod Gebrekrstos Zeru[34,35], Francis Zeukeng[36,37], Stephanie R. M. Zimsen[1], Jonathan F. Mosser[1,11]*

**1** Institute for Health Metrics and Evaluation, University of Washington, Seattle, Washington, United States of America, **2** Department of International Health, Johns Hopkins University, Baltimore, Maryland, United States of America, **3** Department of Microbiology, Immunology and Parasitology, St. Paul's Hospital Millennium Medical College, Addis Ababa, Ethiopia, **4** Department of Microbial, Cellular, and Molecular Biology, Addis Ababa University, Addis Ababa, Ethiopia, **5** Department of Public Health, Hawassa University, Hawassa, Ethiopia, **6** Department of Public Health, Ministry of Health, Hawassa, Ethiopia, **7** Department of Public Health, Jimma University, Jimma, Ethiopia, **8** Department of Public Health, Wachemo University, Hossana, Ethiopia, **9** Institute for Disease Modeling, Bill & Melinda Gates Foundation, Seattle, Washington, United States of America, **10** Big Data Institute, University of Oxford, Oxford, United Kingdom, **11** Department of Health Metrics Sciences, School of Medicine, University of Washington, Seattle, Washington, United States of America, **12** Child Health Analytics Research Program, Telethon Kids Institute, Perth, Western Australia, Australia, **13** Institute of Public Health, University of Gondar, Gondar, Ethiopia, **14** Division of Epidemiology and Biostatistics, The University of Queensland, Brisbane, Queensland, Australia, **15** School of Population Health, Curtin University, Perth, Western Australia, Australia, **16** Department of Community Medicine, University of Ibadan, Ibadan, Nigeria, **17** Department of Community Medicine, University College Hospital, Ibadan, Nigeria, **18** The Malaria Atlas Project, Telethon Kids Institute, Perth, Western Australia, Australia, **19** Geospatial Health and Development Team, Telethon Kids Institute, Perth, Western Australia, Australia, **20** Department of Anatomy, University of Medical Sciences, Ondo, Ondo, Nigeria, **21** Department of Anatomy, Olabisi Onabanjo University, Sagamu, Nigeria, **22** Department of Microbiology and Immunology, Dambi Dollo University, Dembi Dollo, Ghana, **23** Sickle Cell Unit, Ho Teaching Hospital, Ho, Ghana, **24** Faculty of Medicine, University Ferhat Abbas of Setif, Setif, Algeria, **25** Division of Infectious Diseases, University Hospital of Setif, Setif, Algeria, **26** Department of Health Statistics, National Institute for Medical Research, Dar es Salaam, Tanzania, **27** Department of Oral and Maxillofacial Surgery, University College Hospital, Ibadan, Nigeria, **28** Campaign for Health and Neck Cancer Education (CHANCE) Programme, Cephas Health Research Initiative Inc, Ibadan, Nigeria, **29** Department of Public Health, Dambi Dollo University, Dembi Dollo, Ethiopia, **30** Department of Epidemiology, Jimma University, Jimma, Ethiopia, **31** Department of Epidemiology, Stellenbosch University, Cape Town, South Africa, **32** Department of Medicine, Northlands Medical Group, Omuthiya, Namibia, **33** Curtin School of Population Health, Curtin University, Perth, Western Australia, Australia, **34** Department of Statistics, Mekelle University, Mekelle, Ethiopia, **35** Department of Biostatistics, Jimma University, Jimma, Ethiopia, **36** The Biotechnology Centre, University of Yaoundé I, Yaounde, Cameroon, **37** Department of Biochemistry and Molecular Biology, University of Buea, Buea, Cameroon

* jmosser@uw.edu

## Abstract

Eradication and elimination strategies for lymphatic filariasis (LF) primarily rely on multiple rounds of annual mass drug administration (MDA), but also may benefit from

**Data availability statement:** The results dataset has been included in the Supporting Information (S1 Data and S2 Data). The code is publicly available via GIT repository (https://github.com/ihmeuw/lf-malaria-overlap). All input estimates used to produce the results dataset are publicly available as indicted in their respective cited publications and at the following URLs: Lymphatic filariasis prevalence https://doi.org/10.1016/S2214-109X(20)30286-2 https://vizhub.healthdata.org/lbd/lf Insecticide-treated net access https://doi.org/10.1038/s41467-021-23707-7 https://data.malariaatlas.org/maps?layers=Interventions:202106_Africa_Insecticide_Treated_Net_Access Insecticide-treated net use https://doi.org/10.1038/s41467-021-23707-7 https://data.malariaatlas.org/maps?layers=Interventions:202106_Africa_Insecticide_Treated_Net_Use Indoor Residual Spraying https://doi.org/10.1186/s12936-020-03216-6 https://data.malariaatlas.org/maps?layers=Interventions:202106_Africa_IRS_Coverage Malaria *Pf*PR prevalence https://doi.org/10.1016/S0140-6736(19)31097-9 https://data.malariaatlas.org/maps?layers=Malaria:202206_Global_Pf_Parasite_Rate Population estimates https://doi.org/10.1080/20964471.2019.1625151 https://hub.worldpop.org/project/categories?id=3 Shapefile base layer https://espen.afro.who.int/tools-resources/data-query-tools/cartography-database.

**Funding:** EC, TG, CK, JBK, AS, JFM, and JW report support for the present manuscript from the Bill and Melinda Gates Foundation, worktag GR024212. The funders had no role in study design, data collection and analysis, decision to publish, or preparation of the manuscript.

**Competing interests:** I have read the journal's policy and the authors of this manuscript have the following competing interests: EC reports payment or honoraria for lectures, presentations, speakers, bureaus, manuscript writing or educational events from the Royal Holloway University of London, and support for attending meetings and/or travel from the Bill and Melinda Gates Foundation; outside the submitted work. JFM reports grant funding from Gavi and support for attending meetings and/or travel from Bill and Melinda Gates Foundation; outside the submitted work.

vector control interventions conducted by malaria vector control programs. We aim to examine the overlap in LF prevalence and malaria vector control to identify potential gaps in program coverage. We used previously published geospatial estimates of LF prevalence from the Institute for Health Metrics and Evaluation, as well as publicly available insecticide-treated net (ITN) access (proportion of the total population with access to ITNs) and use (proportion of the total population that slept under an ITN) estimates among the total population and malaria *Plasmodium falciparum* parasite rates (*Pf*PR) from the Malaria Atlas Project (MAP). We aggregated the 5x5 km$^2$ estimates of LF prevalence estimates and ITN estimates to the implementation unit (IU) level using fractional aggregation, for 33 LF and malaria-endemic locations in Africa, and then overlaid the IU-level aggregates. In this analysis, ITN coverage was low in areas where LF is common, with 51.7% (90/174) of high-LF-prevalence-IUs having both access and use estimates under 40%. Most (67.8%; 61/90) of these low-ITN-coverage, high-LF-prevalence locations were also categorized as high- or highest-prevalence for malaria by *Pf*PR, suggesting suboptimal ITN coverage even in some malaria-co-endemic locations. Even in IUs with high LF prevalence but low malaria prevalence, almost half (48.2%; 39/81) had high levels of access to ITNs. When accounting for population, however, gaps in ITN access in such areas were evident: more individuals lived in high-LF, low-malaria IUs with low ITN access (8.68 million) than lived in high-LF, low-malaria IUs with high ITN access (6.76 million). These results suggest that relying on current malaria vector control programs alone may not provide sufficient ITN coverage for high LF prevalence areas. Opportunities for coordinated vector control programs in places where LF and malaria prevalence are high but ITN coverage is low – or additional ITN distribution in high-LF, low-malaria locations - should be explored to help achieve elimination goals.

## Author summary

Lymphatic filariasis is a vector-borne disease that can cause significant disability. There is evidence that insecticide-treated nets used by malaria programs can contribute to lymphatic filariasis elimination, but current lymphatic filariasis programs primarily focus on mass drug administration. As funding for programs has stalled and interventions have become more costly, there is a greater interest and need for vector management to be better integrated across sectors and diseases, with WHO promoting integrated vector management specifically for countries co-endemic with LF and malaria. We sought to review the overlap in lymphatic filariasis prevalence and malaria insecticide-treated nets across endemic African countries to identify areas where net distribution can be enhanced. We used previously published, publicly available lymphatic filariasis prevalence and malaria insecticide-treated net coverage results from the Institute for Health Metrics and Evaluation and the Malaria Atlas Project, respectively. Areas with

high lymphatic filariasis prevalence were largely found to have low insecticide-treated net coverage. There is a need for disease programs to work together to maximize effective tools and methods to help achieve elimination goals. The impact of insecticide-treated nets on lymphatic filariasis prevalence will be location-specific and depend on a variety of epidemiological and programmatic factors.

## Introduction

Lymphatic filariasis (LF) is a vector-borne disease caused by the filarial nematodes *Wuchereria bancrofti*, *Brugia malayi*, and *Brugia timori,* and is primarily transmitted by *Anopheles*, *Aedes*, *Culex*, and *Mansonia* mosquito species, varying geographically [1]. LF can lead to permanent disability, including that related to lymphedema and hydrocele, and causes significant mental, social, and financial burden to those afflicted. Under the World Health Organization (WHO)–established Global Programme to Eliminate Lymphatic Filariasis (GPELF), many countries have made significant progress: 17 countries have entered post-validation surveillance (ongoing transmission monitoring following GPELF certification recognizing elimination of LF as a public health problem), 11 have reached post–mass drug administration (MDA) surveillance, and all but two remaining countries have delivered MDA in some capacity [1]. To build upon these gains, the neglected tropical disease (NTD) Roadmap 2021–2030, in alignment with the Sustainable Development Goals, aims to eliminate LF as a public health problem in 58 countries by 2030 [2,3]. In 34 of the countries in the WHO Africa Region and Sudan, LF is a threat to approximately 406 million people [1,4]. In 2019, LF was estimated to have a prevalence rate of 1,472.22 cases per 100,000 (1,024.05 - 2,194.37) and contribute 432,679.92 Disability-adjusted life years (255,366.1 – 729,720.21) for the African Union alone [5]. Within this region, the countries with the highest prevalence include Nigeria, Côte d'Ivoire, the Democratic Republic of the Congo, and Mozambique, which made up approximately 57.6% of the region's prevalent cases in 2019 [5].

Eradication and elimination strategies in endemic African countries primarily rely on multiple rounds of MDA and may additionally benefit from malaria vector control programs, since *Anopheles* species are one of the vectors of LF [6]. Malaria vector control initiatives, particularly insecticide-treated net (ITN) programs, have increased in recent years and contributed to ongoing success in combating malaria, while increasing evidence suggests secondary impacts on other vector-borne diseases [7]. However, although LF and malaria are largely co-endemic, areas with persistently high LF prevalence may not always coincide with areas where malaria prevalence or vector control is high [8,9]. In 2011, WHO released a statement promoting integrated vector management specifically for countries co-endemic with LF and malaria [10]. This statement was followed by the Global Vector Control Response 2017–2030, which aims to reduce mortality and incidence due to vector-borne diseases by at least 75% and 60% respectively, while also preventing epidemics by increasing capacity, enhancing surveillance, and improving coordination and integrated action across diseases and programs [11]. As global funding stalls and the cost of implementing interventions increases [7], it is more important than ever for vector control to be integrated across sectors and disease programs. To enhance cross-disease vector control management, it is crucial to identify where current vector control programs could be expanded to have the most impact. Here we aim to provide one of the first examinations of the overlap in LF prevalence and malaria vector control across endemic Africa to identify potential gaps in program coverage.

## Methods

We used previously published geospatial LF prevalence estimates from the Institute for Health Metrics and Evaluation (IHME) [12] as well as publicly available ITN access and use estimates among the total population, indoor residual spraying (IRS) estimates among the total population, and malaria *Plasmodium falciparum* parasite rates (*Pf*PR) from the Malaria Atlas Project (MAP) [13]. Briefly, the LF estimates were created using Bayesian model-based geostatistics and

time-series methods to generate spatially continuous estimates of global, all-age LF prevalence as measured by immunochromatographic test (ICT) in 2000–2018 [12]. The ITN estimates used a Bayesian mixed modeling framework, and the IRS estimates were generated by collating IRS deployment data from various sources and converting to a standard proportion of households sprayed within the administrative division [13–15]. For *Pf*PR, a cartographic approach was taken for 36 high-burden countries, while a surveillance approach was taken for other *Pf*-endemic countries [16]. Further details regarding the methodology used to create each set of estimates can be found in their respective publications [12–16]. For the purpose of this analysis, we chose to compare the most recent year available for each dataset at the time of analysis to present the most recent comparisons possible: 2018 for LF prevalence, 2019 for malaria prevalence, and 2020 for the vector control datasets. Unlike ITN access and use, which changes very rapidly year-to-year, LF epidemiology and elimination happen on longer time scales [4]. As such areas with high LF prevalence in 2018 are likely to be the same as in 2020, and therefore the maps presented below give the most up-to-date view of this overlap that is currently possible with available results. For a more direct comparison, figures using data from only 2018 have been provided in S4-S10 Figs.

Using population estimates from WorldPop [17], we aggregated the 5x5 km$^2$ estimates of LF prevalence, ITN use, ITN access, malaria prevalence, IRS use, and population to the Expanded Special Project for Elimination of Neglected Tropical Diseases (ESPEN) administrative implementing units (IU) level using fractional aggregation, for 33 LF and malaria-endemic locations in Africa, and then overlaid the IU-level aggregates [18]. IUs represent the administrative units designated by a country to be used for intervention implementation [19]. While these are typically at the district level, there may be variation between countries depending on the structure and objectives of each country's control program [19]. To account for partial coverage of the 5 km$^2$ grid by the IU boundaries and water bodies, we used fractional aggregation, whereby grid cells overlapping multiple IUs were proportionally distributed using the fraction of the cell lying within each IU. Our analyses included a total of 5,195 IUs and 162,868 overlapping 5x5 km$^2$ grids.

We use two ITN metrics in this analysis: access, or the proportion of people among the total population who have access to an ITN; and use, or the proportion of the total population that use an ITN. Following Bertozzi-Villa et al., 2021 [14] we refer to specific metrics, like access and use, by name, and use coverage to more generally refer to combinations of metrics. This analysis focused on the proportion of the total population that had access to ITNs and the proportion of the total population that slept under an ITN, as the use of IRS has largely declined since 2010 [7]. However, maps of LF prevalence and IRS use have been included in S1 and S2 Figs [13]. All maps were created using ESPEN IU shapefiles as the base layer, which are made freely available under the Creative Commons Attribution 4.0 International License (CC BY 4.0) for academic use [18].

In this analysis, we defined IUs having LF prevalence ≥5% as high, while those with a prevalence <5% were considered low. In the absence of well-established standard classification thresholds, we categorized ITN access, use, and malaria prevalence using the following definitions based on the IU-level distributions of these metrics: lowest (<20%); low (20-<40%); high (40%-<60%); highest (≥60%).

## Results

In this analysis, although only 3.4% (174/5,195) of IUs were categorized as having high LF prevalence (Table 1), a total of 38.2 million individuals lived in these locations, primarily located in Nigeria, Côte d'Ivoire, and Liberia. Of those living in these high prevalence areas, 21.9 million (57.3%) lived in IUs with low ITN access, and 1.66 million (4.3%) in IUs with the lowest ITN access, accounting for 51.7% (90/174) and 1.2% (2/174) IUs, respectively (Fig 1). In contrast, there were 10.6 million individuals (27.8%) living in 35.6% (62/174) of IUs with high ITN access and 13.5 million individuals (35.3%) in 42.0% (73/174) of IUs with high ITN use, while only 4.0 million (10.5%) lived in IUs (11.5%; 20/174) with the highest access and 1.8 million people (4.7%) in IUs (6.3%; 11/174) with the highest use (Fig 2).

Just under half of the high LF prevalence locations had at least 40% ITN access (47.1%; 82/174 of IUs) and use (48.3%; 84/174) (Figs 3 and 4). For high LF prevalence areas, 51.7% (90/174) had both ITN access and use in the low or

**Table 1. Table of number of IUs by LF prevalence, ITN access, and ITN use stratified by level of malaria *Pf*PR. LF: Lymphatic filariasis; ITN: Insecticide-treated bednets; *Pf*PR: *Plasmodium falciparum* parasite rate.**

| | | LF prevalence | | | | | | | | All |
|---|---|---|---|---|---|---|---|---|---|---|
| | | Low | | | | High | | | | |
| | | Lowest malaria *Pf*PR | Low malaria *Pf*PR | High malaria *Pf*PR | Highest malaria *Pf*PR | Lowest malaria *Pf*PR | Low malaria *Pf*PR | High malaria *Pf*PR | Highest malaria *Pf*PR | Total |
| **ITN Access** | Lowest | 889 | 115 | 19 | 0 | 0 | 2 | 0 | 0 | 1,025 |
| | Low | 916 | 188 | 82 | 0 | 4 | 24 | 57 | 5 | 1,276 |
| | High | 890 | 695 | 236 | 30 | 6 | 33 | 17 | 6 | 1,913 |
| | Highest | 521 | 323 | 115 | 2 | 1 | 11 | 8 | 0 | 981 |
| **ITN Access Total** | | 3,216 | 1,321 | 452 | 32 | 11 | 70 | 82 | 11 | 5,195 |
| **ITN Use** | Lowest | 1,225 | 130 | 19 | 0 | 2 | 4 | 0 | 0 | 1,380 |
| | Low | 811 | 294 | 104 | 1 | 2 | 21 | 56 | 5 | 1,294 |
| | High | 836 | 673 | 265 | 31 | 6 | 38 | 23 | 6 | 1,878 |
| | Highest | 344 | 224 | 64 | 0 | 1 | 7 | 3 | 0 | 643 |
| **ITN Use Total** | | 3,216 | 1,321 | 452 | 32 | 11 | 70 | 82 | 11 | 5,195 |

LF prevalence 2018 (%) and
ITN access 2020 (%)

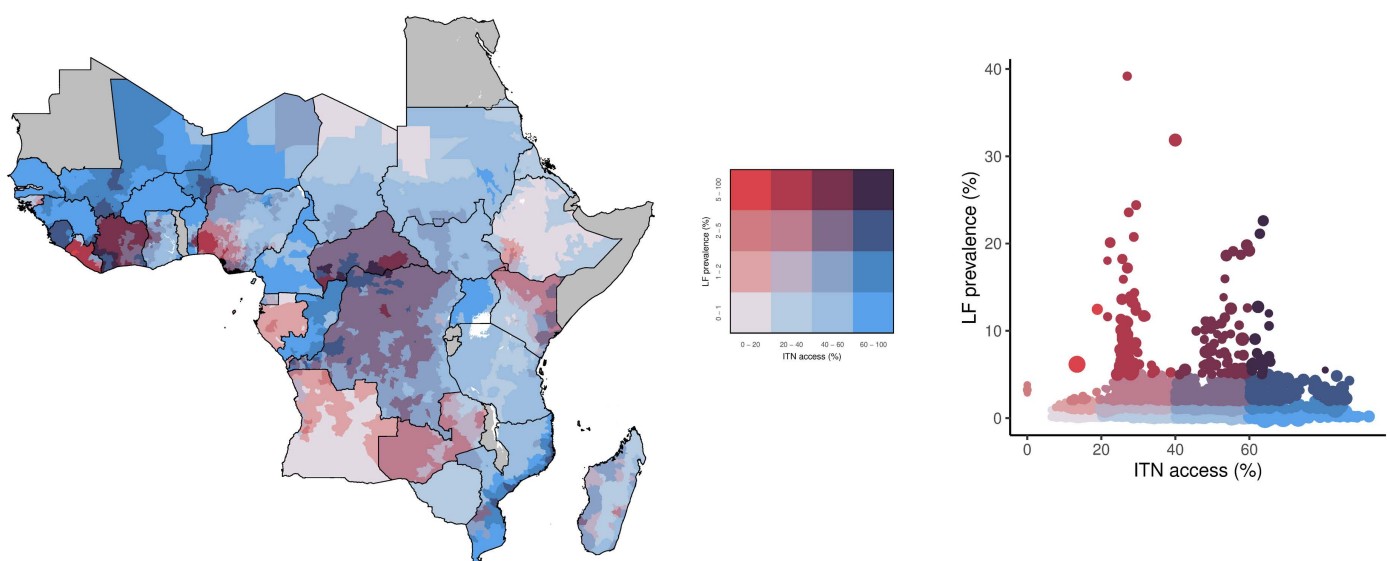

**Fig 1. Overlay map of LF prevalence (counts; 2018) and ITN access among the total population (%; 2020).** The bivariate choropleth map and scatter plot color key in the center show the degree to which LF prevalence (vertical axis, white to red) and ITN access (horizontal axis, white to blue) overlap. Grey indicates areas considered to be non-endemic. LF: lymphatic filariasis; ITN: insecticide-treated net. Map base layer shapefile is from ESPEN, available from: https://espen.afro.who.int/tools-resources/data-query-tools/cartography-database [18].

lowest categories. Geographically, areas with high LF prevalence and low ITN use were concentrated in Liberia, Zambia, Kenya, Angola, and Nigeria, whereas large parts of the Central African Republic, Democratic Republic of the Congo, Côte d'Ivoire, Mali, and Sierra Leone had high LF prevalence and high ITN use (Fig 4).

Approximately 53.5% (93/174) of high LF prevalence areas had high or highest prevalence of malaria, representing 18.4 million (48.2%) individuals. Of these areas, 33.3% (31/93) of IUs had both ITN access and use ≥ 40%, of which only

LF prevalence 2018 (%) and
ITN use 2020 (%)

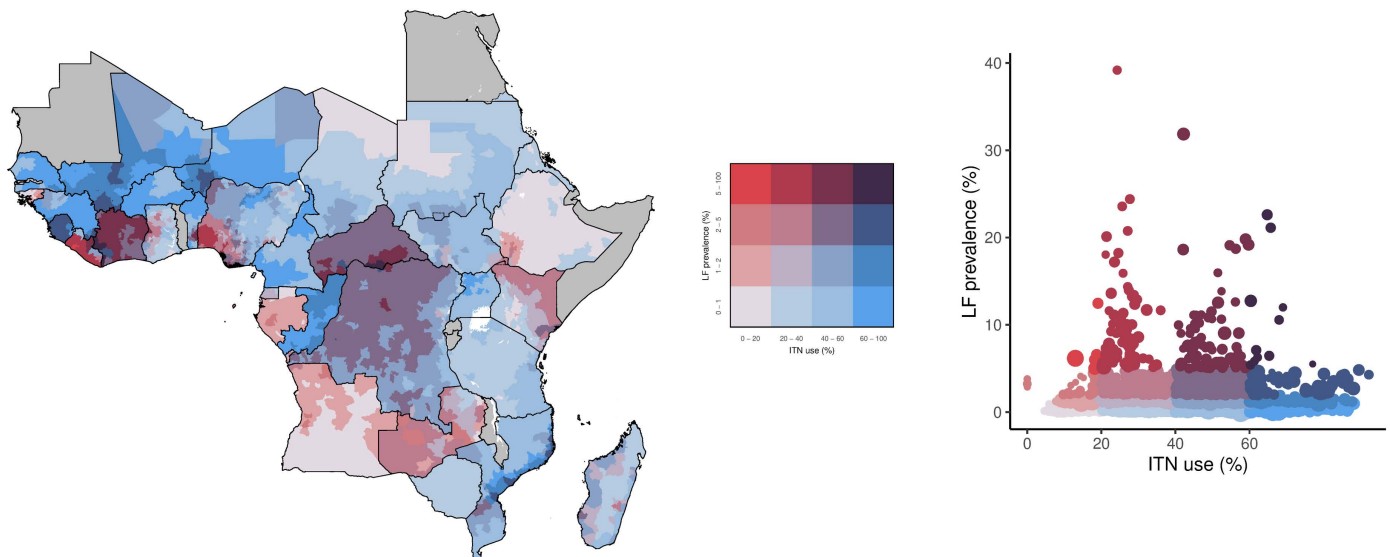

**Fig 2. Overlay map of LF prevalence (counts; 2018) and ITN use among the total population (%; 2020).** The bivariate choropleth map and scatter plot color key indicate the degree to which LF prevalence (vertical axis, white to red) and ITN use (horizontal axis, white to blue) overlap. Grey indicates areas considered to be non-endemic. LF: lymphatic filariasis; ITN: insecticide-treated net. Map base layer shapefile is from ESPEN, available from: https://espen.afro.who.int/tools-resources/data-query-tools/cartography-database [18].

LF prevalence 2018 (%) and
ITN access 2020 (%)

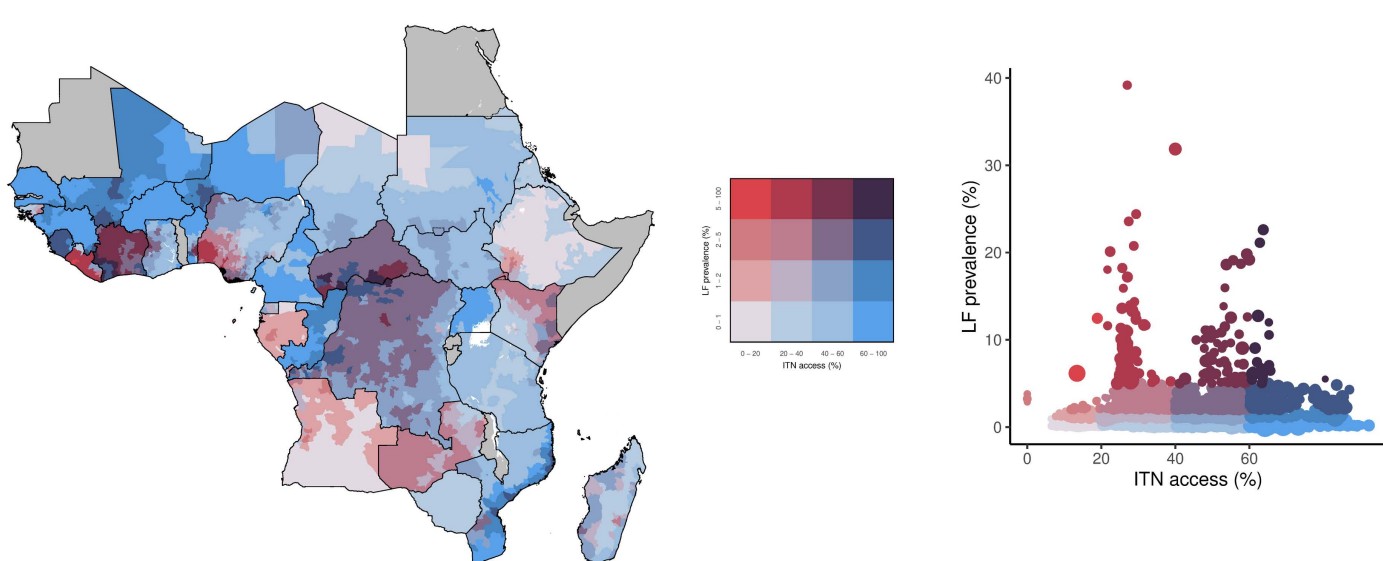

**Fig 3. Overlay map and scatter plot of LF prevalence by (%; 2018) and ITN access among the total population (%; 2020).** The bivariate choropleth map and scatter plot color key in the center indicate the degree to which LF prevalence (vertical axis, white to red) and ITN access (horizontal axis, white to blue) overlap. Grey indicates areas considered to be non-endemic. LF: lymphatic filariasis; ITN: insecticide-treated net. Map base layer shapefile is from ESPEN, available from: https://espen.afro.who.int/tools-resources/data-query-tools/cartography-database [18].

LF prevalence 2018 (%) and
ITN use 2020 (%)

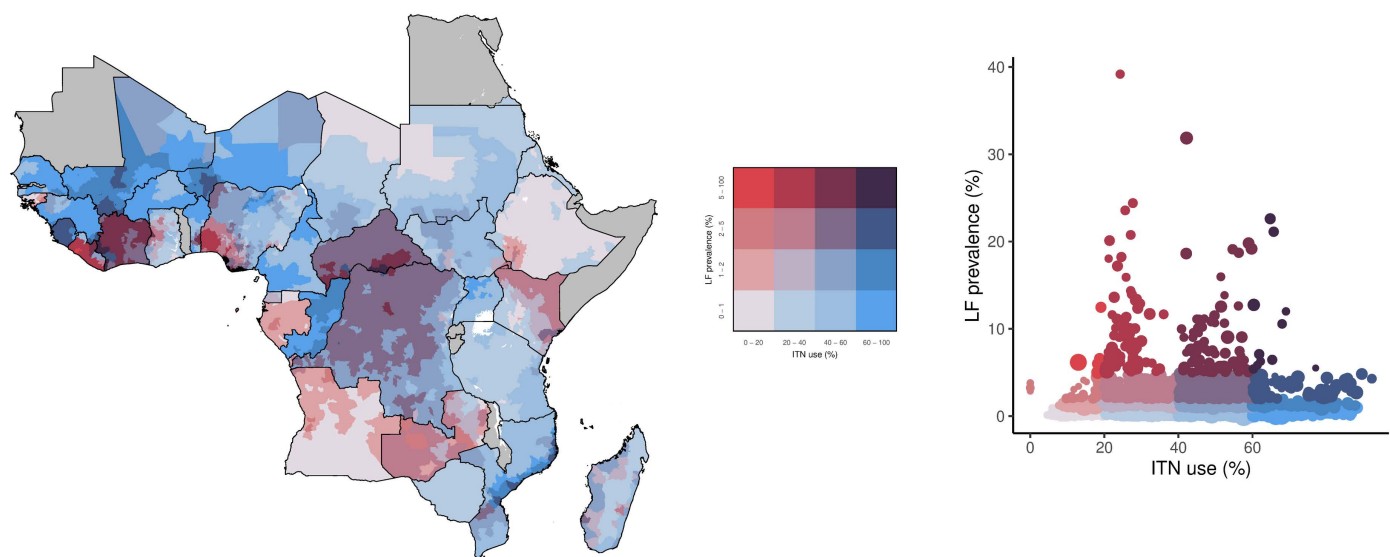

**Fig 4. Overlay map of LF prevalence (%; 2018) and ITN use among the total population (%; 2020).** The bivariate choropleth map and scatter plot color key in the center indicate the degree to which LF prevalence (vertical axis, white to red) and ITN use (horizontal axis, white to blue) overlap. Grey indicates areas considered to be non-endemic. LF: lymphatic filariasis; ITN: insecticide-treated net. Map base layer shapefile is from ESPEN, available from: https://espen.afro.who.int/tools-resources/data-query-tools/cartography-database [18].

9.7% (3/31) had the highest levels of both access and use. IUs with high prevalence of both LF and malaria most commonly had low access (66.7% [62/93] of IUs, containing 13.3 million [72.3%] people) and low use (65.6% [61/93] of IUs, containing 12.7 million [69.0%] people). None of the areas with high prevalence of both LF and malaria had both access and use in the lowest categories, however.

Of the IUs with high LF prevalence but low malaria prevalence, almost half had high ITN access (48.2%; 39/81) and over half had high ITN use (54.3%; 44/81) representing 6.76 million and 8.08 million individuals respectively. Even though only around one third of these IUs (34.6%; 28/81) had low ITN access, there were more people living in these IUs (8.68 million) than in those with high ITN access (6.76 million). These low access IUs were primarily located in Nigeria.

Of the 903 million individuals living in the remaining 96.7% (5,021/5,195) of IUs with low LF prevalence, 200 million (22.2%) lived in IUs with low ITN access (23.6% of IUs; 1,186/5,021) and 114 million (12.6%) in IUs with the lowest ITN access (20.4%; 1,023/5,021). There were 509 million (56.4%) individuals living in IUs with low LF prevalence but with ITN access ≥40% (56.0% of IUs; 2,812/5,021), compared to 124 million people (13.7%) living in IUs with the highest ITN access (19.1%; 961/5,021 IUs). Over half of these low LF prevalence areas had ITN use < 40% (51.5%; 2,584/5,021). Of these, there were fewer areas with low ITN use (24.1%; 1,210/5,021) than the lowest use (27.4%; 1,374/5,021) (Fig 4).

Among all low-LF-prevalence areas, 9.6% (484/5,021) were high-or-highest-prevalence for malaria, accounting for 84.6 million (9.4%) individuals (S3 Fig). Within these, 18.4 million (21.8%) individuals lived in IUs with low ITN use (21.7%; 105/484 IUs) and 14.0 million individuals in IUs with low ITN access (16.9%; 82/484 IUs). Only 3.9% (19/484) of these low-LF prevalence, but high-or-highest malaria prevalence IUs, containing 1.6 million people, had the lowest ITN access and use. In comparison, 50.0% (242/484) of these IUs had high ITN access and use, and 13.2% (64/484) the highest.

## Discussion

This analysis found that most individuals living in high-LF-prevalence areas live in places with low ITN coverage, with the majority falling in the low coverage range. Conversely, the majority of individuals living in areas of low LF prevalence had high ITN coverage. Over half of the areas with high LF prevalence also had high malaria prevalence, but only around one third of these areas had high ITN coverage. For the high LF prevalence areas with low malaria prevalence, despite almost half of IUs having high ITN access, more individuals lived in IUs with low ITN access, suggesting that relying on current malaria vector control programs may not be sufficient for some high LF prevalence areas.

Previous studies evaluating the intersection of LF prevalence, ITN coverage, and malaria prevalence have primarily been limited to a subset of endemic African countries [20–24]. Several of the findings from this analysis, such as ITN coverage being generally low in areas where LF prevalence was high and partial overlap of high LF and malaria prevalence areas, echo the results of these other studies [21,23]. There are likely several factors contributing to the low coverage of ITNs seen in high-LF areas in this analysis. ITN access has been closely linked to development assistance for health funding (DAH), with global organizations playing an important role in deciding resource and program priorities between and within countries [25,26]. Funding for ITNs from organizations such as the Global Fund to Fight AIDS, Tuberculosis and Malaria focuses primarily on high malaria burden areas rather than on areas with high LF prevalence, which likely contributes to some of the low ITN coverage seen for these locations in this analysis [27–29]. Furthermore, many of the countries that initially benefitted from DAH support, much of which was from the Global Fund, tended to be from the same region, such as Eastern Africa [30,31], including some areas where LF prevalence was considered relatively low or non-endemic due to the historical use of dichlorodiphenyltrichloroethane (DDT) spraying against human African trypanosomiasis (HAT) [32]. Importantly, our analysis identified areas with both high malaria prevalence and high LF prevalence but low ITN coverage, suggesting an opportunity for coordinated vector control activities between programs.

In countries affected by war and civil unrest, disruptions to health services and support for ongoing disease programs are likely to further contribute to the observed ITN coverage patterns [33]. Countries also differ in their utilization of ITNs as well as MDA depending on political commitments, competing priorities, and health system structures. In the past, some malaria ITN programs primarily distributed ITNs via antenatal clinics and Expanded Programme for Immunization (EPI) visits, with pregnant persons and children serving as the main target populations due to high health burdens in these groups [34]. For LF, these strategies may have been suboptimal, given that preventing infection across the lifespan is of particular importance to prevent the disabling sequelae of chronic and repeated parasite exposure [35,36]. Although antenatal and EPI visits still play an important role in some countries for continuous ITN distribution [37], since 2007 the WHO has recommended a shift in distribution strategy towards universal coverage [10,37], and more recently towards the subnational tailoring of interventions [38]. In alignment with these recommendations, collaboration to increase the access and use of nets in the highest priority areas could be considered to extend benefits to LF control programs.

The partial overlap seen in this analysis of areas with high LF prevalence and high malaria prevalence could indicate that LF programs may need to consider alternative ITN distribution methods or special net programs for high-LF but low-malaria locations with low ITN coverage, such as some of those outlined in the WHO's document on scaling up ITNs [34]. Despite ITN use among those with nets usually being high [14,39], there are known factors contributing to ongoing ITN non-use, including decreased risk perception during dry seasons for areas where malaria is seasonal [30]. Seasonal trends in LF and malaria may be similar in settings where the same *Anopheles* vectors account for most of the transmission for both diseases [6,8], but due to the chronic nature of LF, these extended periods of ITN non-use could be particularly harmful for those living in LF-endemic areas [4].

While vector control is not currently required for validation of EPHP, there is evidence that it helps to greatly reduce LF prevalence in some settings and could accelerate elimination and eradication programs [8,22,40–46]. A study in Papua New Guinea found that the introduction of ITNs directly led to a decrease in the annual infective biting rate [42], and in

The Gambia, LF elimination was reached in the absence of MDA while scaling up ITNs [47]. However, as LF can be transmitted by multiple vector species, the impact of ITNs on LF prevalence is likely to vary depending on the predominant vector species and their habits, such as whether the species tends to feed outdoors (exophagic) or indoors (endophagic) and when peak biting times occur [8,45,48,49]. Encouragingly, a study conducted in an area of Southeastern Nigeria endemic for both malaria and LF that had not undergone MDA due to co-endemicity with *Loa loa* showed that, even in the presence of multiple LF vectors and high transmission, LF transmission could be halted with the use of ITNs alone – though coverage of 1 net per 2 people in each household was required to do so [46]. Vector control programs also have the potential to drive vector behavior modification which may lead to decreased ITN impact over time [50–52]. Furthermore, location differences in MDA coverage, ITN implementation, IRS use, and other factors will also affect ITN impact.

In Africa, *Anopheles* are the most widespread species, whereas *Culex* and *Mansonia* are more localized in east Africa and west Africa, respectively [53–55]. Elimination of LF by MDA alone may be more likely in areas where *Anopheles* are the primary vector than for other species [6,48,55], but others have argued that adding vector control to MDA in *Anopheles*-dominant areas would be advantageous [20,45,56,57], and WHO specifically recommends the use of ITN in areas where *Anopheles* is the primary vector for LF [4]. This is even more important in urban areas where it is costly and challenging to implement MDA [58], as well as areas that are co-endemic with *Loa loa* where the combination drug approach of MDA drugs for LF (in Africa, albendazole with either ivermectin or diethylcarbamazine), is not recommended, and instead a combination of albendazole-only MDA and vector control is preferred [1].

Differential insecticide resistance patterns should also be considered in areas where the primary vectors are *Anopheles* or *Culex* [59–61] but should not deter the use of ITNs [49,62–65]. Importantly, with the increased concern regarding the spread of the urban-dwelling *Anopheles stephensi* [66], malaria vector control programs may begin shifting from their historically rural focus to include more urban areas, which could increase the potential for overlap between malaria vector control efforts and LF-prevalent locations. Furthermore, *Anopheles stephensi* has been found to coexist with others such as *Aedes* and *Culex* [66–69], presenting an opportunity for broader vector control collaboration to combat not only LF and malaria, but also other mosquito-borne diseases such as dengue.

This analysis carries some limitations. We chose to compare the most recent available geospatial estimates for LF prevalence, ITN coverage, and malaria prevalence, in order to produce the most up-to-date comparisons possible at the time of analysis. As LF results were only available through the year 2018, however, these results do not fully reflect any recent changes in the current spread and level of LF prevalence. Furthermore, both LF and malaria estimates may be subject to accuracy limitations where data is sparse. As mentioned, this analysis used malaria ITN estimates among the total population and does not account for any ITN distribution that may have happened outside of malaria vector control initiatives. An analysis of ITN coverage and LF prevalence over time was outside the scope of this paper, though has been examined in previous country-specific analyses [20].

To the authors' knowledge this is one of the first papers looking at the overlap of LF and ITNs across endemic Africa. These results illustrate the degree to which malaria control programs have achieved access to and use of ITNs in LF-endemic areas. Where the predominant vector species distributions and the context of MDA and other control efforts suggest a role for ITNs in LF control and elimination, these results help to identify locations where additional ITN coverage may be of the most benefit for both diseases. In high-LF, low-malaria locations with low ITN coverage, LF-driven programs to enhance ITN coverage may be needed. Spatial analyses like these can be combined with other context-specific knowledge to help inform future elimination and control strategies.

## Supporting information

**S1 Fig. Overlay map of LF prevalence (%; 2018) and IRS use (%; 2020).** The bivariate choropleth map and scatter plot color key in the center indicate the degree to which LF prevalence (vertical axis, white to red) and IRS use (horizontal axis, white to blue) overlap. Grey indicates areas considered to be non-endemic. LF: lymphatic filariasis; IRS: indoor

residual spraying. Map base layer shapefile is from ESPEN, available from: https://espen.afro.who.int/tools-resources/data-query-tools/cartography-database [18].

(PDF)

**S2 Fig. Overlay map of LF prevalence (counts; 2018) and IRS use (%; 2020).** The bivariate choropleth map and scatter plot color key in the center indicate the degree to which LF prevalence (vertical axis, white to red) and IRS use (horizontal axis, white to blue) overlap. Grey indicates areas considered to be non-endemic. LF: lymphatic filariasis; IRS: indoor residual spraying. Map base layer shapefile is from ESPEN, available from: https://espen.afro.who.int/tools-resources/data-query-tools/cartography-database [18].

(PDF)

**S3 Fig. Overlay map of LF prevalence (%; 2018) and malaria *Pf*PR prevalence (%; 2019).** The bivariate choropleth map and scatter plot color key in the center indicate the degree to which LF prevalence (vertical axis, white to red) and malaria *Pf* prevalence (horizontal axis, white to blue) overlap. Grey indicates areas considered to be non-endemic. LF: lymphatic filariasis; *Pf*: *Plasmodium falciparum.* Map base layer shapefile is from ESPEN, available from: https://espen.afro.who.int/tools-resources/data-query-tools/cartography-database [18].

(PDF)

**S4 Fig. Overlay map of LF prevalence (%; 2018) and ITN access among the total population (%; 2018).** The bivariate choropleth map and scatter plot color key in the center indicate the degree to which LF prevalence (vertical axis, white to red) and ITN access (horizontal axis, white to blue) overlap. Grey indicates areas considered to be non-endemic. LF: lymphatic filariasis; ITN: insecticide-treated nets. Map base layer shapefile is from ESPEN, available from: https://espen.afro.who.int/tools-resources/data-query-tools/cartography-database [18].

(PDF)

**S5 Fig. Overlay map of LF prevalence (counts; 2018) and ITN access among the total population (%; 2018).** The bivariate choropleth map and scatter plot color key in the center indicate the degree to which LF prevalence (vertical axis, white to red) and ITN access (horizontal axis, white to blue) overlap. Grey indicates areas considered to be non-endemic. LF: lymphatic filariasis; ITN: insecticide-treated nets. Map base layer shapefile is from ESPEN, available from: https://espen.afro.who.int/tools-resources/data-query-tools/cartography-database [18].

(PDF)

**S6 Fig. Overlay map of LF prevalence (%; 2018) and ITN use among the total population (%; 2018).** The bivariate choropleth map and scatter plot color key in the center indicate the degree to which LF prevalence (vertical axis, white to red) and ITN use (horizontal axis, white to blue) overlap. Grey indicates areas considered to be non-endemic. LF: lymphatic filariasis; ITN: insecticide-treated nets. Map base layer shapefile is from ESPEN, available from: https://espen.afro.who.int/tools-resources/data-query-tools/cartography-database [18].

(PDF)

**S7 Fig. Overlay map of LF prevalence (counts; 2018) and ITN use among the total population (%; 2018).** The bivariate choropleth map and scatter plot color key in the center indicate the degree to which LF prevalence (vertical axis, white to red) and ITN use (horizontal axis, white to blue) overlap. Grey indicates areas considered to be non-endemic. LF: lymphatic filariasis; ITN: insecticide-treated nets. Map base layer shapefile is from ESPEN, available from: https://espen.afro.who.int/tools-resources/data-query-tools/cartography-database [18].

(PDF)

**S8 Fig. Overlay map of LF prevalence (%; 2018) and IRS use (%; 2018).** The bivariate choropleth map and scatter plot color key in the center indicate the degree to which LF prevalence (vertical axis, white to red) and IRS use (horizontal

axis, white to blue) overlap. Grey indicates areas considered to be non-endemic. LF: lymphatic filariasis; IRS: indoor residual spraying. Map base layer shapefile is from ESPEN, available from: https://espen.afro.who.int/tools-resources/data-query-tools/cartography-database [18].
(PDF)

**S9 Fig. Overlay map of LF prevalence (counts; 2018) and IRS use (%; 2018).** The bivariate choropleth map and scatter plot color key in the center indicate the degree to which LF prevalence (vertical axis, white to red) and IRS use (horizontal axis, white to blue) overlap. Grey indicates areas considered to be non-endemic. LF: lymphatic filariasis; IRS: indoor residual spraying. Map base layer shapefile is from ESPEN, available from: https://espen.afro.who.int/tools-resources/data-query-tools/cartography-database [18].
(PDF)

**S10 Fig. Overlay map of LF prevalence (%; 2018) and malaria *Pf*PR prevalence (%; 2018).** The bivariate choropleth map and scatter plot color key in the center indicate the degree to which LF prevalence (vertical axis, white to red) and malaria *Pf* prevalence (horizontal axis, white to blue) overlap. Grey indicates areas considered to be non-endemic. LF: lymphatic filariasis; *Pf*: *Plasmodium falciparum*. Map base layer shapefile is from ESPEN, available from: https://espen.afro.who.int/tools-resources/data-query-tools/cartography-database [18].
(PDF)

**S1 Data. Codebook.**
(XLSX)

**S2 Data. Results dataset.**
(CSV)

## Author contributions

**Conceptualization:** Joanna L. Whisnant, Mustafa Kamal Sikder, Jonathan F. Mosser.

**Data curation:** Joanna L. Whisnant, Mustafa Kamal Sikder, Tsegaye Alemu, Amelia Bertozzi-Villa, Annie J. Browne, Paulina Agnieszka Dzianach, Yalemzewod Assefa Gelaw, Peter W. Gething, Taren M. Gorman, Simon I. Hay, Olayinka Stephen Ilesanmi, Juniper Boroka Kiss, Jailos Lubinda, Michael A. McPhail, Gideon Olamilekan Oluwatunase, Amel Ouyahia, Susan Fred Rumisha, Adam Saddler, Afeez Abolarinwa Salami, Francesca Sanna, Daniel J. Weiss, Naod Gebrekrstos Zeru, Francis Zeukeng, Stephanie R. M. Zimsen.

**Formal analysis:** Joanna L. Whisnant, Mustafa Kamal Sikder, Amelia Bertozzi-Villa, Annie J. Browne, Paulina Agnieszka Dzianach, Yalemzewod Assefa Gelaw, Juniper Boroka Kiss, Jailos Lubinda, Michael A. McPhail, Susan Fred Rumisha, Adam Saddler, Francesca Sanna, Daniel J. Weiss, Jonathan F. Mosser.

**Investigation:** Mustafa Kamal Sikder.

**Methodology:** Joanna L. Whisnant, Mustafa Kamal Sikder, Tsegaye Alemu, Mubarek Yesse Ashemo, Paulina Agnieszka Dzianach, Peter W. Gething, Taren M. Gorman, Simon I. Hay, Michael A. McPhail, Susan Fred Rumisha, Adam Saddler, Jonathan F. Mosser.

**Project administration:** Olivia D. Nesbit, Jonathan F. Mosser.

**Supervision:** Simon I. Hay, Jonathan F. Mosser.

**Validation:** Joanna L. Whisnant, Mustafa Kamal Sikder, Gizachew Taddesse Akalu, Tsegaye Alemu, Mubarek Yesse Ashemo, Amelia Bertozzi-Villa, Ewerton Cousin, Paulina Agnieszka Dzianach, Simon I. Hay, Olayinka Stephen Ilesanmi, Cathleen Keller, Michael A McPhail, Olivia D. Nesbit, Verner N. Orish, Amel Ouyahia, Susan Fred Rumisha,

Desalegn Shiferaw, Jacques Lukenze Tamuzi, Naod Gebrekrstos Zeru, Francis Zeukeng, Stephanie R. M. Zimsen, Jonathan F. Mosser.

**Visualization:** Joanna L. Whisnant, Mustafa Kamal Sikder, Afeez Abolarinwa Salami.

**Writing – original draft:** Joanna L. Whisnant, Mustafa Kamal Sikder, Amelia Bertozzi-Villa, Ewerton Cousin, Paulina Agnieszka Dzianach, Peter W. Gething, Cathleen Keller, Olivia D Nesbit, Daniel J. Weiss, Stephanie R. M. Zimsen, Jonathan F. Mosser.

**Writing – review & editing:** Joanna L. Whisnant, Mustafa Kamal Sikder, Gizachew Taddesse Akalu, Tsegaye Alemu, Amelia Bertozzi-Villa, Annie J. Browne, Ewerton Cousin, Simon I Hay, Olayinka Stephen Ilesanmi, Cathleen Keller, Olivia D. Nesbit, Gideon Olamilekan Oluwatunase, Verner N. Orish, Amel Ouyahia, Afeez Abolarinwa Salami, Jacques Lukenze Tamuzi, Francis Zeukeng, Stephanie R. M. Zimsen, Jonathan F. Mosser.

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
