## [Decision Letter · Decision Letter 0]

Examining the overlap in lymphatic filariasis prevalence and malaria insecticide-treated net access-use in endemic Africa

Dear Dr. Whisnant,

Thank you for submitting your manuscript to PLOS Neglected Tropical Diseases. After careful consideration, we feel that it has merit but does not fully meet PLOS Neglected Tropical Diseases's publication criteria as it currently stands. Therefore, we invite you to submit a revised version of the manuscript that addresses the points raised during the review process.

Please submit your revised manuscript within 60 days Feb 20 2025 11:59PM. If you will need more time than this to complete your revisions, please reply to this message or contact the journal office at plosntds@plos.org. Please include the following items when submitting your revised manuscript:

We look forward to receiving your revised manuscript.

Kind regards,

Nigel Beebe, PhD

Section Editor

Nigel Beebe

Section Editor

Shaden Kamhawi

co-Editor-in-Chief

Paul Brindley

co-Editor-in-Chief

**Additional Editor Comments :**

Please respond to reviewers comments and suggestions, especially reviewer 2.

**Journal Requirements:**

1) Please upload all main figures as separate Figure files in .tif or .eps format. For more information about how to convert and format your figure files please see our guidelines: 

2) Some material included in your submission may be copyrighted. According to PLOSu2019s copyright policy, authors who use figures or other material (e.g., graphics, clipart, maps) from another author or copyright holder must demonstrate or obtain permission to publish this material under the Creative Commons Attribution 4.0 International (CC BY 4.0) License used by PLOS journals. Please closely review the details of PLOSu2019s copyright requirements here: PLOS Licenses and Copyright. If you need to request permissions from a copyright holder, you may use PLOS's Copyright Content Permission form.

Potential Copyright Issues:

i) Figures 1, 2, 3, 4, and S1-S10. Please (a) provide a direct link to the base layer of the map (i.e., the country or region border shape) and ensure this is also included in the figure legend; and (b) provide a link to the terms of use / license information for the base layer image or shapefile. We cannot publish proprietary or copyrighted maps (e.g. Google Maps, Mapquest) and the terms of use for your map base layer must be compatible with our CC BY 4.0 license.

3) Thank you for stating that "the code is publicly available via GIT repository." Please note that your Data Availability Statement is currently missing the DOI/accession number of each dataset OR a direct link to access each dataset. If your manuscript is accepted for publication, you will be asked to provide these details on a very short timeline. We therefore suggest that you provide this information now, though we will not hold up the peer review process if you are unable.

4) Please amend your detailed Financial Disclosure statement. This is published with the article. It must therefore be completed in full sentences and contain the exact wording you wish to be published.

**Reviewers' Comments:**

Reviewer's Responses to Questions

**Key Review Criteria Required for Acceptance?**

**Methods**

-Are the objectives of the study clearly articulated with a clear testable hypothesis stated?

-Is the study design appropriate to address the stated objectives?

-Is the population clearly described and appropriate for the hypothesis being tested?

-Is the sample size sufficient to ensure adequate power to address the hypothesis being tested?

-Were correct statistical analysis used to support conclusions?

-Are there concerns about ethical or regulatory requirements being met?

Reviewer #1: I am unable to assess the statistical and geospatial analysis in the paper. Other aspects of the methodology are appropriately described.

Reviewer #2: The objectives are clearly stated. The paper aims to overlay published geospatial estimates of ITN access and use, and LF prevalence estimates at 33 LF and malaria endemic locations in Africa. The paper then looks at the ITN coverage in high and low LF prevalence areas to assess how effective malaria control programs have been in providing access and use to ITNs that would potentially contribute to interruption of LF. The study design is appropriate to address this since these spatial datasets are available through the extensive prior published work on the three aspects studied (ITNs access and use, LF prevalence spatial prediction, Malaria prevalence spatial prediction PfPR) that has been done by this group and many others, together with the decades of work of surveyors and national programmes who provided the raw data to the spatial modelers of the three aspects. I appreciated being provided with the prior publications of these parameters as well as IRS coverage. Putting these three predictions together is a valid and useful approach to visualizing and assessing the overlap in ITN access-use and LF/malaria prevalence. The explanation for which year has been chosen for the overlap is clear and reasonable (lines 136 -137). However, there are some gaps in 1) how the results are explained 2) evaluating the implications of the findings for future control. There is no specific hypothesis about what would have been expected as far as I could tell.

Given that the LF prevalence estimates are produced by 5 km2 grid (Local Burden of Disease collaborators, 2019) like the ITN and PfPR, why did you choose to aggregate the estimates into Implementation units (line 147)? What is the usual size of an IU (district or other unit?) and how much does it vary by country?

**Results**

-Does the analysis presented match the analysis plan?

-Are the results clearly and completely presented?

-Are the figures (Tables, Images) of sufficient quality for clarity?

Reviewer #1: Yes.

Reviewer #2: Presentation of results is sometimes very confusing. Definitions need to be very clear and cutoffs (e.g. low' or high' ITN "coverage") justified.

The first issue I had is with the definition of ITN 'coverage'. It is unfortunate that this word has been used to refer to a composite of access and use "where use among the total population and access among the total population were both separately above or below a specified threshold" line 160. ITN coverage was traditionally defined at the household level by WHO and in DHS surveys ("Proportion of HH with at least 1 net"). I understand that this is likely now out of date, as is the household rather than individual definition of access, and better measures such as nets per capita are becoming more common (as defined clearly in the Bertozzi-Villa 2021 paper). Nevertheless it might have been better to coin another term rather than 'coverage' to avoid confusion. Also, a composite measure of two parameters seems less than optimal when one (use) depends on the other (access). I understand the desire for one measure, but this one seems overcomplicated to me.

I find the following sentences in the abstract very obscure and contradictory:

"in the analysis, almost half of the locations (47.1%; 82/174 of IUs) with high LF prevalence (>5%) had at least 40% coverage with ITN access and use. Among high LF prevalence areas, both access and use were low, with 51.7% (90/174) having both access and use estimates under 40%. Additionally when classified using malaria PfPR, most (67.8%; 61/90) of these low ITN coverage, high LF prevalence locations were also considered high prevalence for malaria, Among areas with low LF prevalence (<5%), over half had ITN access >=40% (56.0%, 2812/5021) while only 48.5% (2437/5021) had use >=40%"

I have read this several times, but still cannot grasp what you are trying to say. The problems include

1. sometimes reporting combined access and use, and sometimes the two separately

2. including the % cutoffs for the levels of "coverage" or prevalence (not really necessary once defined in text), next to the % in each category

3. some comparisons using IUs and some using 5km2 units.. why?

So what do we conclude? The malaria programs have successfully covered the LF high prevalence areas, or not? Do we need special net programmes in low malaria/high LF areas?

It might help to provide some tables on the distribution of the geographical units into different categories of LF prevalence and ITN access/use, stratified by malaria PfPR.

I don't see why the LF prevalence count maps are useful (figs 1 and 2) in addition to the LF prevalence maps, if the counts are not adjusted for population. What do they add?

In the Discussion, the first paragraph is very important, and should be captured better in abstract somehow. But I am puzzled by the last sentence (and lines 248 to 250 and lines 259 to 261 which make the same point). If half of the areas had high malaria prevalence, then wouldn't they be targeted for ITN distribution anyway?. If they have low ITN access and use, doesn't that represent a failure of the malaria programme, not need for integration or extra programmes? The areas of most concern would seem to me to be those with low malaria but high LF, where special effort to provide nets for LF might be needed.

**Conclusions**

-Are the conclusions supported by the data presented?

-Are the limitations of analysis clearly described?

-Do the authors discuss how these data can be helpful to advance our understanding of the topic under study?

-Is public health relevance addressed?

Reviewer #1: Yes

Reviewer #2: The last three sentences of the abstract are very general, and could have been written before the work was done. They do not summarize the findings of the study. What can we conclude based on the data presented here? What is the degree to which malaria control programmes have achieved access and use to ITNs in malaria endemic areas? What more should programmes be doing? In which areas should net distribution be enhanced (line 79?).

If net access/use is low in high malaria and LF areas, isn't that a failure of net distribution for malaria, not a need for 'integration' whatever that means.

**Editorial and Data Presentation Modifications?**

Reviewer #1: I am not sure if the addition of published manuscripts as part of the supplementary materials is important. The reference to these manuscripts should suffice.

Reviewer #2: Lines 94 to 96 are not quite correct. Post-validation surveillance (PVS) is not the period of transmission assessment after MDA - that is post MDA surveillance, which is first. PVS comes after validation (several years after MDA stops and multiple TAS surveys have been done)

A couple of places with missing refs line 195 and 225

Results text 163-171, 190-195 and 201 onwards, consider putting into Tables to make clearer. It's quite hard to read at the moment.

The Discussion seems overlong, while the Results section could be expanded to be clearer. The limitations of the data analysis are well described, but a bit belaboured since the work is quite comprehensive and impressive.

**Summary and General Comments**

Reviewer #1: The paper describes the overlap of LF, malaria, and ITN (access and use) and is important in understanding the role of vector control for LF. Although I am unable to assess the statistical approaches, the results are informative. Overall, the paper is well written. There are just some few comments that I believe the authors should address.

Line 55: Italicise Plasmodium falciparum

Line 121: delete the word "endemic" before Africa.

Lines 157 - 158: "For ITN coverage (access and use) and malaria prevalence we used the following definitions: lowest <20%; low 20-39.9%; high 40%-59.9%; highest ≥60%" Kindly clarify. Are these thresholds for ITN coverage or malaria prevalence?

Line 167: Check the reference error

Line 195: Chech the reference error

Line 225: Check the reference error

- An important question to be addressed in this work is the ITN coverage level that results in significant impact on LF prevalence and control. ITN access and use >40% does not really explain the level at which ITNs start having an impact. In Figures 1 and 2, for instance, areas with high LF and high ITN access are shown. However, if high ITN access and use was effective, one would expect low LF prevalence in these high ITN access and use areas. How do the authors explain these?

Lines 273 - 283: In parts of Africa, especially along the East Africa coast, Culex species play a role in the transmission of LF. The authors should explain the overlap of high LF and high malaria in these areas and possibly the low impact of ITNs.

- It is also important to note that malaria vector interventions result in vector behavior modifications, with vectors biting more outdoors than indoors. This could possibly explain the lack of correlation between high malaria and high ITN use and high LF prevalence and should be discussed.

Reviewer #2: Overall, this is a good piece of work combining these datasets but it needs more clarity in the results and more interpretation of what the findings mean for programmes.

PLOS authors have the option to publish the peer review history of their article (what does this mean? ). If published, this will include your full peer review and any attached files.

**Do you want your identity to be public for this peer review?** For information about this choice, including consent withdrawal, please see our Privacy Policy .

Reviewer #1: No

Reviewer #2: No

**Figure resubmission:**

**Reproducibility:**



---

## [Decision Letter · Decision Letter 1]

Response to Reviewers
Revised Manuscript with Track Changes
Manuscript

Shaden Kamhawi

co-Editor-in-Chief

Paul Brindley

co-Editor-in-Chief

**Additional Editor Comments:**
**Journal Requirements:**

**Reviewers' comments:**

**Key Review Criteria Required for Acceptance?**

**Methods**

-Are the objectives of the study clearly articulated with a clear testable hypothesis stated?

-Is the study design appropriate to address the stated objectives?

-Is the population clearly described and appropriate for the hypothesis being tested?

-Is the sample size sufficient to ensure adequate power to address the hypothesis being tested?

-Were correct statistical analysis used to support conclusions?

-Are there concerns about ethical or regulatory requirements being met?

Reviewer #1: Yes

Reviewer #2: Thanks for the diligence in responding to the comments. I am happy with the responses relevant to this section.

**Results**

-Does the analysis presented match the analysis plan?

-Are the results clearly and completely presented?

-Are the figures (Tables, Images) of sufficient quality for clarity?

Reviewer #1: Yes

Reviewer #2: Thanks for providing the new Table classifying IUs as suggested. It is very useful to be able to conceptualize the scale of the problem and should help program managers. I strongly suggest that it be put into the main text as most people don't look at Supplementary files and this table is very important and useful.

**Conclusions**

-Are the conclusions supported by the data presented?

-Are the limitations of analysis clearly described?

-Do the authors discuss how these data can be helpful to advance our understanding of the topic under study?

-Is public health relevance addressed?

Reviewer #1: Yes

Reviewer #2: Revised text is much better and clearer.

**Editorial and Data Presentation Modifications?**

Reviewer #1: (No Response)

Reviewer #2: Ref 19 is a bit garbled (Organization, WH Filariasis GP to EL?). Can be solved in EndNote etc by where you put the comma. WHO should probably be listed as author in several other refs e.g. 1, 2,7, 10, 11, 34, 57.

Something missing ref 53? Biology and control of....

**Summary and General Comments**

Reviewer #1: My comments are appropriately addressed.

Reviewer #2: Overall the authors have been very responsive to the comments and the paper seems much improved as a result.

I have two remaining comments, in addition to suggesting that the new Supp Table be put in main text:

Lines 271-281 about net distribution policies, alluded to a couple of times in the responses: It is true that in the early days of LLIN, nets were prioritized to children under 5 and pregnant women through antenatal clinics. This happened quite a long time ago (>10 yrs) and I am not aware of anywhere that uses this as the sole method now as it is known not to achieve high enough access (it is used a supplementary continuous distribution method in some countries). So I would change this text to refer to past distribution policies, which may indeed have resulted in low access, but policies have already moved on in WHO and other guidelines towards universal (all ages in population) coverage (could be cited https://www.who.int/publications/i/item/guidelines-for-malaria) and followed by Global Fund etc . Gaps now are probably more about lack of sub-national prioritization of universal coverage in highest risk areas rather than blanket national policies. You could say "PAST sub-optimal ITN coverage" and "IN THE PAST, some malaria ITN programs primarily distributed ITNs via antenatal clinics". "Extend the scope of ITN programmes" is a bit vague and might be better as "increase the access and use of nets in the highest priority areas" or similar.

In the paragraph lines 287 to 297 you do not currently cite the paper by Richards et al 2013 Community-Wide Distribution of Long-Lasting Insecticidal Nets Can Halt Transmission of Lymphatic Filariasis in Southeastern Nigeria

https://doi.org/10.4269/ajtmh.12-0775. This is a very relevant paper you could consider citing that shows the impact of LLIN on LF transmission in an area in Africa, with multiple LF vectors, that had not had MDA for LF, comparing full access of nets with targeted (under five/pregnant women) distribution. An important finding apart from the impact on LF transmission was how many nets had to be given out to get full access (1 net per 2 people) in many large households. Disclosure: I co-managed this project and analysis, and am an author on the paper .

The other responses are comprehensive and satisfactory, thanks.

PLOS authors have the option to publish the peer review history of their article (what does this mean? ). If published, this will include your full peer review and any attached files.

**Do you want your identity to be public for this peer review?** For information about this choice, including consent withdrawal, please see our Privacy Policy .

Reviewer #1: No

Reviewer #2: **Yes: ** Patricia M Graves

**Figure resubmission:****Reproducibility:** To enhance the reproducibility of your results, we recommend that authors of applicable studies deposit laboratory protocols in protocols.io, where a protocol can be assigned its own identifier (DOI) such that it can be cited independently in the future. Additionally, PLOS ONE offers an option to publish peer-reviewed clinical study protocols. Read more information on sharing protocols at https://plos.org/protocols?utm_medium=editorial-email&utm_source=authorletters&utm_campaign=protocols

---

## [Decision Letter · Decision Letter 2]

Dear Ms. Whisnant,

We are pleased to inform you that your manuscript 'Examining the overlap in lymphatic filariasis prevalence and malaria insecticide-treated net access-use in endemic Africa' has been provisionally accepted for publication in PLOS Neglected Tropical Diseases.

Best regards,

Nigel Beebe, PhD

Section Editor

Nigel Beebe

Section Editor

Shaden Kamhawi

co-Editor-in-Chief

Paul Brindley

co-Editor-in-Chief

Reviewer's Responses to Questions

**Key Review Criteria Required for Acceptance?**

**Methods**

-Are the objectives of the study clearly articulated with a clear testable hypothesis stated?

-Is the study design appropriate to address the stated objectives?

-Is the population clearly described and appropriate for the hypothesis being tested?

-Is the sample size sufficient to ensure adequate power to address the hypothesis being tested?

-Were correct statistical analysis used to support conclusions?

-Are there concerns about ethical or regulatory requirements being met?

Reviewer #2: Yes

**Results**

-Does the analysis presented match the analysis plan?

-Are the results clearly and completely presented?

-Are the figures (Tables, Images) of sufficient quality for clarity?

Reviewer #2: Yes

**Conclusions**

-Are the conclusions supported by the data presented?

-Are the limitations of analysis clearly described?

-Do the authors discuss how these data can be helpful to advance our understanding of the topic under study?

-Is public health relevance addressed?

Reviewer #2: Yes

**Editorial and Data Presentation Modifications?**

Reviewer #2: I suggest spelling out EPHP (presumably Elimination as a Public Health Problem) in line 294. It's only used once, and some readers may not be familiar with it.

**Summary and General Comments**

Reviewer #2: Thanks for making the changes suggested. All looks good now.

PLOS authors have the option to publish the peer review history of their article (what does this mean? ). If published, this will include your full peer review and any attached files.

**Do you want your identity to be public for this peer review?** For information about this choice, including consent withdrawal, please see our Privacy Policy .

Reviewer #2: **Yes: ** Patricia Graves

---

## [Editor Report · Acceptance letter]

Dear Ms. Whisnant,

We are delighted to inform you that your manuscript, "Examining the overlap in lymphatic filariasis prevalence and malaria insecticide-treated net access-use in endemic Africa," has been formally accepted for publication in PLOS Neglected Tropical Diseases.

Best regards,

Shaden Kamhawi

co-Editor-in-Chief

Paul Brindley

co-Editor-in-Chief
